# A New Generation of Gene Therapies as the Future of Wet AMD Treatment

**DOI:** 10.3390/ijms25042386

**Published:** 2024-02-17

**Authors:** Janusz Blasiak, Elzbieta Pawlowska, Justyna Ciupińska, Marcin Derwich, Joanna Szczepanska, Kai Kaarniranta

**Affiliations:** 1Faculty of Medicine, Collegium Medicum, Mazovian Academy in Plock, 09-402 Plock, Poland; 2Department of Pediatric Dentistry, Medical University of Lodz, 92-217 Lodz, Poland; elzbieta.pawlowska@umed.lodz.pl (E.P.); marcin.derwich@umed.lodz.pl (M.D.); joanna.szczepanska@umed.lodz.pl (J.S.); 3Clinical Department of Infectious Diseases and Hepatology, H. Bieganski Hospital, 91-347 Lodz, Poland; jciupinska@bieganski.com.pl; 4Department of Ophthalmology, University of Eastern Finland, 70210 Kuopio, Finland; kai.kaarniranta@uef.fi; 5Department of Ophthalmology, Kuopio University Hospital, 70210 Kuopio, Finland

**Keywords:** age-related macular degeneration, AMD, vascular endothelial growth factor, *VEGF*, anti-VEGF therapy, gene therapy, clinical trials, gene editing

## Abstract

Age-related macular degeneration (AMD) is an eye disease and the most common cause of vision loss in the Western World. In its advanced stage, AMD occurs in two clinically distinguished forms, dry and wet, but only wet AMD is treatable. However, the treatment based on repeated injections with vascular endothelial growth factor A (VEGFA) antagonists may at best stop the disease progression and prevent or delay vision loss but without an improvement of visual dysfunction. Moreover, it is a serious mental and financial burden for patients and may be linked with some complications. The recent first success of intravitreal gene therapy with ADVM-022, which transformed retinal cells to continuous production of aflibercept, a VEGF antagonist, after a single injection, has opened a revolutionary perspective in wet AMD treatment. Promising results obtained so far in other ongoing clinical trials support this perspective. In this narrative/hypothesis review, we present basic information on wet AMD pathogenesis and treatment, the concept of gene therapy in retinal diseases, update evidence on completed and ongoing clinical trials with gene therapy for wet AMD, and perspectives on the progress to the clinic of “one and done” therapy for wet AMD to replace a lifetime of injections. Gene editing targeting the *VEGFA* gene is also presented as another gene therapy strategy to improve wet AMD management.

## 1. Introduction

Age-related macular degeneration (AMD) is an eye disease and a serious problem in aging societies [1]. Advanced AMD presents two clinically distinct forms: dry (atrophic) and wet (exudative, neovascular) [2]. Both forms may lead to legal blindness and sight loss, but currently, only wet AMD is treatable by intravitreal injections with antibodies to vascular endothelial growth factor A (*VEGFA*) and its receptor (anti-*VEGFA* therapy) [3,4]. However, usually wet AMD treatment does not result in a cure for the disease, but at best stops its progression, preventing or delaying sight loss. In addition, the treatment is troublesome for patients as it includes repeated injections into one eye or both eyes and constitutes a serious financial burden for the patients. An alternative for resistance to or intolerance of anti-*VEGFA* treatment is photodynamic therapy, which may be applied along with the anti-*VEGFA* treatment [5,6].

Although anti-*VEGFA* treatment is a targeted therapy, it is not free from detrimental side effects that may lead to damage to the eye. Some dietary interventions may improve these effects and the efficacy of anti-*VEGFA* therapy, but the effects depend on many factors that may be difficult to anticipate [7]. Moreover, some patients display resistance to this kind of therapy [8].

In 2008, three independent research groups reported a successful subretinal injection of an AAV-based expression vector carrying the retinoid isomerohydrolase *RPE65* (retinal pigment epithelium-specific 65 KDa protein) gene to improve vision in individuals with inherited blindness (Leber’s congenital amaurosis) [9,10,11]. This resulted in the first FDA-approved gene therapy product for the eye—Luxturna (voretigene neparvovec-rzyl). However, possibly more important was that this initial success of gene therapy in the eye contributed to the rejuvenation of gene therapy in general after its serious setback associated with the death of a patient inspired a deeper look into the biology of virus vectors [12,13]. These experiments set the groundwork for the development of gene therapies for AMD.

Good and bad experiences of gene therapy enabled the elaboration of the strategy of safe and efficient cargo delivery by the vectors derived from adeno-associated viruses [14]. Consequently, recent studies on gene therapy in wet AMD have revolutionized the perspective of the treatment of this disease [15]. Instead of a replacement of faulty or lacking protein with its functional counterpart, new therapies direct the eye to synthesize anti-VEGF drugs. These therapies eliminate the burden of multiple intravitreal injections, offering a stable production of anti-VEGF proteins for a long time. The results obtained so far in ongoing and completed clinical trials are promising, but some aspects concerning safety and efficacy need further studies (https://classic.clinicaltrials.gov/ct2/results?cond=Wet+Age-related+Macular+Degeneration&term=VEGF+gene+therapy&cntry=&state=&city=&dist=&Search=Search&type=Intr, accessed on 15 January 2024).

There are excellent reviews addressing gene therapy and gene editing in eye diseases in detail, e.g., [15,16]. In this narrative/perspective review, we update information on ongoing and completed clinical trials on gene therapy for wet AMD. We briefly describe wet AMD pathogenesis and therapy and provide general information on gene therapy in ocular diseases, focusing on the AAV vector, as the main platform to deliver target DNA to retinal cells in wet AMD. Gene editing in wet AMD is also briefly presented as a further perspective in the management of this disease.

## 2. Wet Age-Related Macular Degeneration: Pathogenesis and Therapy

Age-related macular degeneration is a complex, multifactorial disease with aging, environmental/lifestyle, and genetic/epigenetic factors playing a role in its pathogenesis [17]. The complexity of this disease is also underlined in that each of these elements has several variants that may interplay both within each group of factors and with factors from a different group. For instance, AMD is not a monogenic disease, and as shown in genome-wide association studies, several genetic loci may be involved in its pathogenesis, containing genes of the following three main pathways: the complement pathway, lipid metabolism, and extracellular matrix remodeling. Variants of these genes may interact and their effect can be modulated by environmental/lifestyle AMD risk factors, including an unhealthy diet [18]. The most consistently reported loci that are associated with AMD are the rs1061170 (Tyr402His/p.Y402H) single nucleotide polymorphism variant in the complement factor H (*CHF*) and the age-related maculopathy susceptibility 2 and high-temperature requirement A serine peptidase 1 (*ARMS2/HTRA1*) [19,20]. Therefore, many mechanisms may be involved in AMD pathogenesis which, along with the limited possibility of studying live human eyes, results in limited treatment options for AMD.

Advanced AMD happens in two clinically distinct forms, dry and wet, and individuals affected by either form may suffer from gaps or dark spots in their vision (Figure 1). Although wet AMD is responsible for a minority of all AMD cases, it accounts for about 90% of sight loss related to AMD and, somewhat paradoxically, only wet AMD is treatable [21]. The causal relationship, if any, between these two forms of AMD is not clear, and some studies suggest that they might be considered as two distinct diseases [22]. We showed that wet AMD might correlate with mortality in a 12-year prospective case–control study, but the question of whether wet AMD might be an independent risk factor for death is still open [23].

Drusen, small, light, solid objects containing proteins and lipids, are present in both forms of AMD but their presence in dry AMD is much more common than in wet AMD (approximately 90:10%). They can be divided into hard and soft drusen [24]. Dry AMD is characterized by a gradual deterioration of the retina following from the death of retinal cells that are not renewed [25]. Drusen and edema may be visible in a microscopic picture of the affected eye.

Choroidal neovascularization (CVN), presenting the growth of new blood vessels derived from the choroid through a break in the Bruch’s membrane, is a hallmark of wet AMD [26]. In wet AMD the retinal pigment epithelium (RPE) is not broken, as it is in dry AMD, but RPE may be detached [27]. Furthermore, the functionality of the RPE cells is changed as they overproduce *VEGFA* and other angiogenic factors that are essential in CNV. Fresh blood vessels formed in CNV are fragile and release their content into the retina, resulting in fibrosis, with subsequent formation of disciform scar that can result in sight loss unless treated [28]. The angiogenic effect of *VEGFA* in CNV is mainly mediated by the PLCγ-PKC-MAPK pathway initiated from the 1175-PY site on *VEGFA* receptor 2 (VEGFR-2) located on the exterior of endothelium [29].

At present, the treatment of patients with antibodies against *VEGFA* and its receptor is the only accepted therapy in wet AMD, whose efficacy has been consistently confirmed and has become a routine procedure in ophthalmic clinics [30]. The anti-*VEGFA* treatment significantly improves the outcomes of wet AMD, although cases of improving visual acuity are rare; the treatment stops the progression of the disease and ameliorates damage in the retina [3,31]. However, the treatment per se is troublesome, as it involves repeated intravitreal injections, every 8 to 12 weeks after three-monthly loading doses, with a relatively expensive formulation of monoclonal antibodies against *VEGFA* and its receptor.

Many formulations of *VEGFA* antibodies are present in the market, including the following that are FDA approved: aflibercept (Eylea), brolucizumab (Beovu), ranibizumab (Lucentis), and faricimab-svoa (Vabysmo); bevacizumab (Avastin) is used off-label as it is primarily designated as an anti-cancer drug [32]. Also, biosimilars to anti-VEGF have been recently approved by FDA (https://www.reviewofophthalmology.com/article/an-update-on-the-antivegf-biosimilar-pipeline; accessed on 15 January 2024).

In summary, the introduction of anti-*VEGFA* treatment revolutionized the therapy for wet AMD. Although the treatment does not cure the disease, it may protect against sight loss and is in general safe for repeated applications over a long period of time. However, the necessity of lifetime intravitreal injections and the relatively high cost of anti-VEGF drugs are a serious burden for patients.

## 3. Gene Therapy for Wet Age-Related Macular Degeneration

The eye seems to be predisposed as a target for gene therapy due to its relatively small size, with a compartmentalized structure and immune-privileged status [16]. These eye characteristics lower the risk of systemic exposure. Moreover, advanced non-invasive methods of imaging in the eye, including optical coherence tomography, fundoscopy, angiography, and two-photon microscopy, assist in the real-time monitoring of the progress of the gene therapy procedures and their safety [33]. Another advantageous feature of the eye for gene therapy is that changes in a single gene may be associated with various clinical states. For example, homozygous mutation in the “historic” *RPE65* gene may result in either Leber’s congenital amaurosis 2 or rare forms of retinitis pigmentosa (RP) [34].

The development of gene therapy has brought a better understanding of the biology of viral vectors, as they are basic vectors used in eye gene therapy [35]. In general, virus vectors can be divided into integrating and non-integrating with the host genome. Due to safety concerns, non-integrating viral vectors may be the current strategies and those used in at least the near future for transgene delivery in gene therapy [36].

Among many non-integrating viral vector types, adeno-associated viral (AAV) vectors have many features of key significance for eye gene therapy [37].

Adeno-associated viruses consist of two parts, an icosahedral protein capsid and a single-stranded DNA (ssDNA) genome [38,39,40]. The AAV genome has 4.8 kb of ssDNA and two T-shaped inverted terminal repeats (ITRs), on either end of the genome [41] (Figure 2). The terminal repeats flank two open reading frames, *Rep* and *Cap*, that are transcribed and translated to produce the virus life cycle proteins Rep78, Rep68, Rep52, Rep40, and VP1, VP2, and VP3 capsid proteins resulting from the use of alternate promoters and alternate splicing.

In the transfer plasmid construct, which is to be packaged by the AAV virus, the transgene is placed between ITRs, and *Rep* and *Cap* are attached in trans (Figure 2).

AAV presents various serotypes that differ in the capsid structure, immunogenicity, and cellular tropism; the most suitable serotypes can be chosen to accommodate the eye environment [42]. The parental adeno-associated virus requires a helper virus to replicate to exert its pathogenic action in humans [43]. Moreover, AAV vectors were shown to elicit a limited immune response due to the restricted ability of the parent virus to infect antigen-presenting cells [44].

The virus needs the Rep protein to replicate and the open reading frame for encoding this protein can be removed by gene editing, depriving the virus of the ability to replicate and integrate [43]. If this open reading frame is not edited, the virus integrates in a site-specific fashion with the human genome on chromosome 19 [45]. Therefore, the AAV vector offers the possibility to integrate or not with the host genome by the manipulation with the *Rep* locus.

The AAV vectors have some limitations [46]. They cannot package fragment DNA longer than 4.5 kb, their long-term expression is limited to non-dividing cells and, despite their limited ability to elicit immune response, there is a high incidence of pre-existing immunity in humans [47].

If the *Rep* locus is retained, one may expect a long-term, controlled expression of target DNA in chromosome 19 and accumulation of the product of that expression. Therefore, if the target DNA could encode a product that would stimulate the eye to continuously produce an antagonist of *VEGFA*, this would replace multiple injections with anti-*VEGFA* antibodies. This is the general concept of the exploitation of AAV vectors in gene therapy of wet AMD. This is a definite change of the general strategy of gene therapy which, in its standard form, aims to replace a faulty or absent gene with a gene that displays normal expression. Here, nothing is replaced, but the strategy with the AAV vector carries an added value for target cells. Moreover, the recent strategies of gene therapies with the AAV vectors have created the possibility of therapeutic gene manipulations with minimal concerns for inflammatory and immune reactions and with maximal efficacy [16].

The above idea is just an idea, and many details need to be elaborated before its accomplishment. The main problems to solve are the delivery route and sustained expression of the *VEGFA* antagonist.

In general, three routes of the delivery of a transgene are considered in ocular gene therapy delivery methods (Figure 3). The delivery to the suprachoroidal and subretinal spaces is characterized by targeted access and wide transduction of the retinal cells, low exposure to the vitreous and anterior segments, and compartmentalized delivery of the AAV vector as observed in preclinical studies [48,49,50]. Subretinal delivery poses a low risk of immune response and inflammation [51]. Intravitreal injections are characterized by limited transduction of retinal cells, a wide exposure of the vitreous and anterior segments, and a high risk of immune/inflammatory reactions [52,53].

Subretinal introduction of the *RPE65* gene in patients with Leber’s congenital dystrophy was a fundamental piece of work for gene therapy for AMD. The normal *RPE65* gene encodes the *RPE65* protein that transforms all-trans-retinol into 11-cis-retinol in the retinal cycle, a reaction that is crucial for the conversion of visual information into electric pulses [54]. Transduction of RPE cells with the transgene results in the production of the functional *RPE65* protein. Therefore, experiments showed that AAV vectors may be used to provide a stable expression of the target DNA in the retina, opening a perspective for the “one-and-done” in-office treatment of retinal diseases requiring a continuous production of a therapeutic protein. However, the question remains whether a subretinal injection is an optimal route of administration of the target DNA.

It appears likely that the past, present, and future of gene therapy in the eye is not limited to the use of AAV-based vectors as delivery systems, and the use of several other types of viruses, as well as non-viral vectors, has been attempted. More detailed analysis can be found elsewhere, e.g., in the review of Rodrigues et al. [47].

## 4. Clinical Trials on Gene Therapy for Wet Age-Related Macular Degeneration

ClinicalTrials.gov lists 22 studies on gene therapy in wet AMD (https://clinicaltrials.gov/search?cond=Wet%20AMD&intr=gene%20therapy, accessed on 16 January 2024). These include one completed with results, four completed, twelve recruiting, three active, not recruiting, one not yet recruiting, and one enrolling by invitation studies. In this section, we provide some brief details on two clinical trials, ADVM-022 and RGX-314, as they present the most advanced stages of such studies. In general, these studies aim to provide a durable expression of therapeutic levels of intraocular protein and to maintain vision, to safely reduce the current treatment burden.

### 4.1. ADVM-022

The ADVM-022 open-label clinical trial (OPTIC) by Adverum Biotechnologies was designed to elaborate an in-office intravitreal gene therapy for wet AMD and diabetic macular edema (https://clinicaltrials.gov/study/NCT03748784, accessed on 15 January 2024).

The preclinical studies of the OPTIC trial were performed in mice with laser-induced CNV [55]. Those studies utilized the AAV2.7m8 capsid, which was obtained from AAV2 by directed evolution and optimizing the efficacy of transduction of the retinal cells after intravitreal administration [56]. Therefore intravitreal injection was chosen in that study as not only the vitreous body is easily accessible, but it also avoids possible barriers for target DNA delivery resulting from dense tissue penetration, while guaranteeing a widespread delivery to the outer retina [56]. The results of several other studies contributed to the initiation and success of the project. These include studies on the stabilization of the expression of aflibercept. It was shown that a single injection of ADVM-022 in non-human primates resulted in a 30-day sustained expression of aflibercept, and an injection was effective in CVN inhibition, even if performed 13 months before laser-induction of CNV [57,58].

The expressional cassette of ADVM-022 contained the aflibercept cDNA, which was codon-optimized, following the translation start codon within the Kozak consensus sequence (Figure 4) [55]. The expression of the cassette was driven from the human cytomegalovirus promoter, and several other regulatory elements optimized the expression of aflibercept, which is a recombinant chimeric protein consisting of parts of the human *VEGFA* binding portion of *VEGFA* receptors and human igG1 immunoglobulin [59].

The main general aim of ADVM-022 was to treat wet AMD with a single intravitreal injection of the AAV2.7m8 capsid containing cDNA for an alfibercept-like protein. The Phase I trial enrolled patients who were previously controlled with frequent anti-VEGF injections and who received high or low doses of ADVM-022, which was renamed as ixoberogene soroparvovec (ixo-vec). The high dose population showed a 98% decrease in the number of anti-VEFGA injections required, while the low dose group showed an 80% reduction [60]. Ocular treatment-emergent adverse events (TEAEs) related to ixo-vec were mild to moderate and included anterior chamber cells and vitreal cells. Five serious TEAEs (SAEs) occurring during the trial, including two cases of cataract, dry AMD, retinal detachment, and a case of recurrent uveitis. However, dry AMD and recurrent uveitis could not have been attributed to ixo-vec treatment. No clinical or imaging evidence of inflammation was reported, but some patients showed the presence of anterior chamber cells and vitreal cells. Best-corrected visual acuity was stable in both groups over 2 years. The mean change in central subfield thickness decreased after ixo-vec injection and remained stable for over 2 years.

In summary, the ADVM-022 trial showed that a single injection of ixo-vec induced sustained levels of intraocular aflibercept, ameliorating retinal anatomy, no worsening of BCVA, and a significant reduction in the number of anti-VEGF injections, with a majority of patients not needing such supplementation of the ixo-vec treatment. The main limitation of this trial was a small sample size, which in combination with some baseline characteristics might have caused a bias. The study was unmasked, creating a potential for unintentional decisions made by investigators. There was no comparison between ixo-vec treatment and cases of wet AMD that were not treated. Despite these and other limitations, the ADVM-022 has opened a perspective to abandon frequent anti-*VEGFA* injection in place of just a single injection of ixo-vec.

### 4.2. RGX-314

RGX-314 is a clinical trial also aiming at the strategy of a one-time treatment for wet AMD with an estimated enrollment of 465 participants (https://classic.clinicaltrials.gov/ct2/show/NCT05407636, accessed on 16 January 2024). The trial uses an AAV8 vector carrying the Fab fragment that is similar to ranibizumab, a humanized antibody to *VEGFA*.

The preclinical study in non-human primates (NHPs) showed that mRNA and protein after subretinal delivery of anti-VEGF Fab-containing AAV8 vector were distributed widely throughout the retina [61]. A retinal demelanization was observed in all injection sites, with no intraocular inflammation. The low dose of the vector induced a difference in retinal thickness, whereas the high dose resulted in about a 10% decrease in total retinal thickness at the injection site. Some dose-dependent histopathological changes at 3 months were observed, including low-grade inflammation in the choroid, retina, and sclera. That study showed the presence of anti-VEGF Fab in the anterior chamber at 30 days in all RGX-314 doses and its feasibility to define an upper dose limit in further escalation studies.

Clinical trials renamed the vector to ABBV-RGX-314, likely due to the sponsor. Currently, ATMOSPHERE (NCT04704921), and ASCENT (NCT05407636), are two pivotal, active, and enrolling patient trials evaluating the subretinal delivery of ABBV-RGX-314 in individuals with wet AMD. In addition, AAVIATE (NCT04514653) is a Phase II trial for the treatment of wet AMD using suprachoroidal delivery of ABBV-RGX-314.

Phase I/IIa of RGX-314 is complete and long-term follow-up continues. It has demonstrated that RGX-314 delivered subretinally via a transvitreal approach is generally well-tolerated with no abnormal immune response, or ocular inflammation (http://www.prnewswire.com/news-releases/regenxbio-announces-additional-positive-interim-phase-iiia-and-long-term-follow-up-dataof-rgx-314-for-the-treatment-of-wet-amd-301228344.html, accessed on 18 January 2024). Although as many as 20 SAEs were reported in 13 patients, only one, a significant decrease in vision, was possibly drug-related.

Phase II of the ABBV-RGX-314 pharmacodynamic study with subretinal delivery of the vector was conducted to evaluate the clinical performance of ABBV-RGX-314 from the planned commercial process (BRX) versus the initial clinical research process (Hyperstack, HS) (https://www.regenxbio.com/science-innovation/publications/; accessed on 19 January 2024). Both processes showed a similar clinical profile. The primary endpoint was 6 months after the subretinal delivery of ABBV-RGX-314. The ABBV-RGX-314 protein levels were similar in cohorts administered both high and low doses of the drug. An improvement in BCVA between three and eight letters was observed. The fraction of injection-free subjects was 73%. The study reported five SAEs in four patients, and none were considered drug-related. Common adverse effects were related to surgery and included conjunctival hemorrhage and inflammation.

Phase II of the AAVIATE study with suprachoroidal delivery of investigational ABBV-RGX-314 for wet AMD aimed to evaluate the mean change in BCVA for ABBV-RGX-314 compared with ranibizumab monthly injections, at month 9 (https://www.regenxbio.com/science-innovation/publications/; accessed on 19 January 2024). That study observed 15 SAEs but none was considered drug-related. No cases of chorioretinal vasculitis occlusion or hypotony were observed. Individual TEAEs cases included intraocular inflammation, conjunctival hemorrhage, increase in intraocular pressure, conjunctival hyperemia, and episcleritis. BCVA worsened after ABBV-RGX-314 injection, compared with ranibizumab.

In summary, the RGX-314 program includes clinical trials and preclinical studies showing that the AAV serotype 8 vector may be useful in both subretinal and subchoroidal injection of plasmid encoding an anti-VEGF protein and supporting its stable expression, resulting in a meaningful percentage of patients that no longer require anti-VEGF injections. Although some safety issues should be addressed, the procedure is generally safe and there are very few, if any, drug-related SAEs. The subretinal route of administration shows some advantages over the suprachoroidal route, but the clinical trials are ongoing and we expect to learn more about the efficacy and safety of RGX-314 soon.

### 4.3. Trials with Anti-VEGFA Treatment-Naive Patients

Completed and ongoing clinical trials have mostly recruited patients who responded to anti-*VEGFA* therapy, and the number of injections during the trial and before it may be one of the trial endpoints. It is difficult to enroll patients who would refuse a standard anti-*VEGFA* therapy in favour of a possibly more uncertain gene therapy, without knowledge about its outcome.

The Phase 1, multicenter, open-label study to assess the efficacy and safety of two doses of the adeno-associated viral vector serotype 2 (AAVCAGsCD59) expressing sCD59 (CD59 molecule (CD59 blood group), soluble form) administered via intravitreal injection seven days after a single intravitreal injection of anti-VEGF is enrolling patients with treatment-naive wet AMD, but no results have been posted thus far (https://clinicaltrials.gov/study/NCT03585556?cond=Wet%20AMD&term=gene%20therapy&rank=4, accessed on 19 January 2024). Another ongoing trial, entitled “A Study of Genetic and Environmental Factors and Their Effect on Response to Treatment With Lucentis (Ranibizumab) for Wet AMD” has also recruited patients who have never been treated with any anti-*VEGFA* drug, but no results have been posted (https://clinicaltrials.gov/study/NCT00469352?cond=Wet%20AMD&term=gene%20therapy&rank=10#participation-criteria accessed on 19 January 2024). The trial “Comparison of Treatment Regimens Using Ranibizumab: Intensive (Resolution of Intra- and Sub-retinal Fluid) vs. Relaxed (Resolution of Intra-retinal Fluid and/or Sub-retinal Fluid > 200 µm at the Foveal Centre) (FLUID)” lists as an exclusion criterion “treatment with any anti-angiogenic drugs (including any anti-VEGF agents) before baseline in the studied eye (allowed in the other eye) (https://clinicaltrials.gov/study/NCT01972789?cond=Wet%20AMD&term=gene%20therapy&page=2&rank=19#participation-criteria, accessed on 19 January 2024)”. This study does not compare responders to anti-*VEGFA* therapy with patients who are naive to such therapy.

The anti-VEGF responders experience a treatment burden imposed by frequent injections. Therefore, trials with anti-*VEGFA* naive wet AMD patients may be a perspective challenge.

## 5. Genome Editing and Wet AMD

Genome editing, as its name suggests, is a method to make specific changes within the genome. The straightforward application of genome editing is the correction of mutations that are causative for inherited diseases [62]. Clustered regularly interspaced palindromic repeats (CRISPR)/Cas9 is currently the most common platform to edit the nuclear genome, with promising results for its mitochondrial counterpart [63,64,65]. However, CRISPR/Cas9 has several drawbacks, including genotoxicity and limited control of outcomes [66]. That is why some modifications to the classical CRISPR-Cas system have been successfully introduced in the treatment of inherited retina diseases [67,68,69,70,71,72,73].

Base editors are CRISPR-Cas-based systems, allowing for a targeted conversion of a single base pair that can be replaced by or supplemented with prime editing, which is a versatile tool enabling the introduction of all types of transition/transversion as well as small deletions and insertions [74,75]. Adenine base editors were used to restore visual function in mice with a de novo mutation in the *RPE65* gene [72]. The expression of the base editors occurred after the subretinal injection of a lentivirus-based vector.

The most spectacular success of prime editing in eye disease therapy was restoring visual functions in a mouse model of retinitis pigmentosa through correcting the inherited missense mutation in exon 13 of the phosphodiesterase 6B (*PDE6B*) gene, whose product is essential for the initiation of rod phototransduction [76,77,78]. The packaging system used in that experiment was based on the AAV virus.

Although ClinicalTrials.gov does not list any items under the entry “wet AMD genome editing”, the NCT06031727 “CRISPR/cas13-medIated RNA tarGeting tHerapy for the Treatment of Neovascular Age-related Macular Degeneration Investigator-initiated Trial (SIGHT-I)” is a clinical trial on genome editing in wet AMD (https://clinicaltrials.gov/study/NCT06031727?cond=Wet%20AMD&term=gene%20theapy&page=3&rank=26; accessed on 18 January 2024).

Intravitreal delivery of a Cas9-based system in an AAV serotype 9 vector to knockout the *VEGFA* gene to a laser-induced wet AMD mouse model resulted in a decreased level of the *VEGFA* protein in RPE and reduced neovascularization in the choroid [79]. That experiment showed a 45% reduction in the CNV area, while aflibercept injection decreased the area by 39%. Moreover, unlike aflibercept, this genome editing strategy ensured a long-lasting therapeutic effect with a single injection. Similar results were obtained in several other studies that also used the subretinal route of vector delivery [80,81,82]. Several studies have addressed the application of genome editing technologies to knockout the *VEGFA* gene and inhibit CNV in various research schemes (reviewed in [16]).

Even classical gene therapy, replacing a faulty gene with its normal copy, is a kind of genome editing or at least an attempt to do so. Genome editing with CRISPR/Cas 9, base editing, and prime editing represent more precise tools than classical gene therapy and open much broader perspectives for regulating phenotype by manipulation of the genome.

## 6. Conclusions, Outstanding Questions, and Perspectives

The introduction of anti-*VEGFA* treatment in wet AMD revolutionized the therapy of this earlier incurable disease, but this treatment, in most cases, does not cure wet AMD as it mainly stops the progression of the disease, preventing or delaying sight loss. Moreover, such therapy, which is characterized by the lifetime intravitreal injections every 8 to 12 weeks after three monthly loading doses with relatively expensive substances, constitutes a serious burden for patients. Therefore, attempts to replace repeated injections with a single dose of a therapeutic are justified and they are a hope for a more comfortable life for millions of wet AMD patients [83].

Patients’ comfort should be considered in planning a therapeutic strategy, but the main points to consider are safety and efficacy. It should be stressed that anti-*VEGFA* treatment in wet AMD cannot a priori be completely efficient, as *VEGFA* is the key regulator of normal and pathological angiogenesis, but it is not the only factor responsible for neovascularization in wet AMD [84,85].

A new generation of gene therapies opens a perspective for a future “one and done” in-office treatment for wet AMD, thus replacing a lifetime of injections. This is still theoretical, as today we can only say that the continuous release of anti-*VEGFA* substance at a therapeutic level might last for up to 3 years after a single injection of a recombinant DNA. Therefore, an outstanding question is how long can gene therapy-induced expression of a therapeutic protein last. Answering this question requires future prospective and retrospective studies, but even if 3 months is the longest injection-free period, it would be a substantial improvement for wet AMD patients, even if a fraction of them require additional injections.

The next outstanding question is how far in the future does the treatment last. This question is difficult to answer based on completed and ongoing clinical trials, as real-world patients usually have worse outcomes than those achieved in clinical trials [86]. Phase 3 trials are in the planning or recruiting stages and it is estimated that if these trials show results comparable with corresponding 1/2 trials, the new “one-and-done” in-office therapy for wet AMD may be available in 3–4 years (https://www.aao.org/eyenet/article/gene-therapy-for-amd, accessed on 16 January 2024).

Several other outstanding questions may be asked concerning future trials and preclinical studies. In general, trials of genome therapy in wet AMD do not show a breakthrough in the efficacy of the treatment and the main outcome is a reduction of the number of injections to reach a therapeutic effect. Therefore, the trials that recruited anti-*VEGFA* treatment naive patients may achieve results that are not influenced by the burden resulting from previous injections, and so they seem to be a better platform for searching for a mode of more efficient anti-*VEGFA* therapy.

Promising results obtained in genome editing in retinal diseases and the regulation of *VEGFA* justify further research in this area. However, as mentioned, *VEGFA* is not the only protein that regulates angiogenesis, and it seems that gene editing creates more possibilities to act on more than one angiogenic protein than traditional gene therapy [87].

The use of vectors as vehicles to deliver therapeutic constructs to the target site and the delivery route are the most important issues in planning any gene therapy strategy [88]. Adeno-associated viruses have an established position in therapeutic genetic manipulation in the retina [89]. However, the use of AAV vectors in the retina is not free from some obstacles, which are analyzed in a recent review by Zin [90]. They include limited tropism, penetration of the retinal inner limiting membrane, small packaging capacity, neutralizing antibody/immune response, and others. However, current studies on the delivery of cargo in gene therapy in the retina focus on the improvement of the AAV vectors, rather than replacing them with others. Therefore, optimizing the construct of an AAV vector for retinal genetic manipulation is a challenge, both in traditional gene therapy and genome editing in the retina and so in wet AMD.

Retinal pigment epithelium is a key structure in AMD pathogenesis, which is especially observed in the dry form of the disease, featured by a partial loss of RPE cells. In wet AMD, the breakdown of the blood–retinal barrier and RPE dysfunction cause the massive release of *VEGFA* and neoangiogenesis [91]. Therefore, it must be stressed that new generations of gene therapies reviewed here are a treatment for the consequences and not the causes of wet AMD, and so they are not an ultimate therapy for wet AMD. A further perspective of wet AMD therapy should include AMD-related changes in RPE cells, and so it might be applicable to both forms of AMD. However, this is a future perspective, and at present several concerns still require addressing in relation to a new generation of gene therapy.

In summary, a new generation of gene therapies in wet AMD, including genome editing, opens a perspective of a significant improvement of the lifetime comfort of wet AMD patients. Over a 3–4-year period, these may replace multiple anti-*VEGFA* injections with a single dose administered in-office.

## Figures and Tables

**Figure 1 ijms-25-02386-f001:**
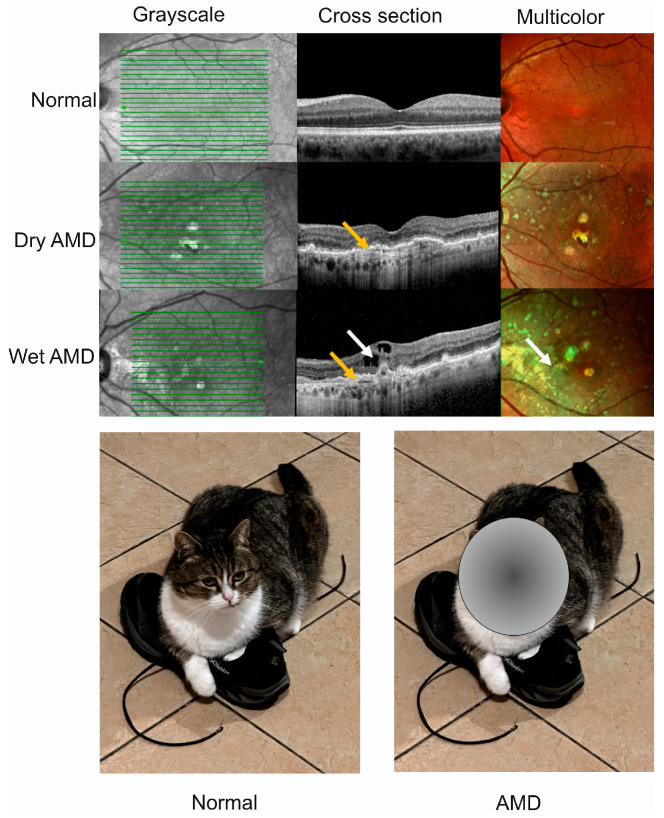
Advanced age-related macular degeneration (AMD) presents two clinically distinct forms: dry and wet. Optical coherence tomography of normal, dry AMD, and wet AMD eyes (**upper panel**). Yellow arrows indicate degenerated retinal pigment epithelial layers (green lines) and a white arrow shows edema and intraretinal fluid observed in cross-sectional view corresponding to increased intensity of green color in the multicolor image. Advanced AMD causes disturbances in or loss of central vision (**lower panel**).

**Figure 2 ijms-25-02386-f002:**
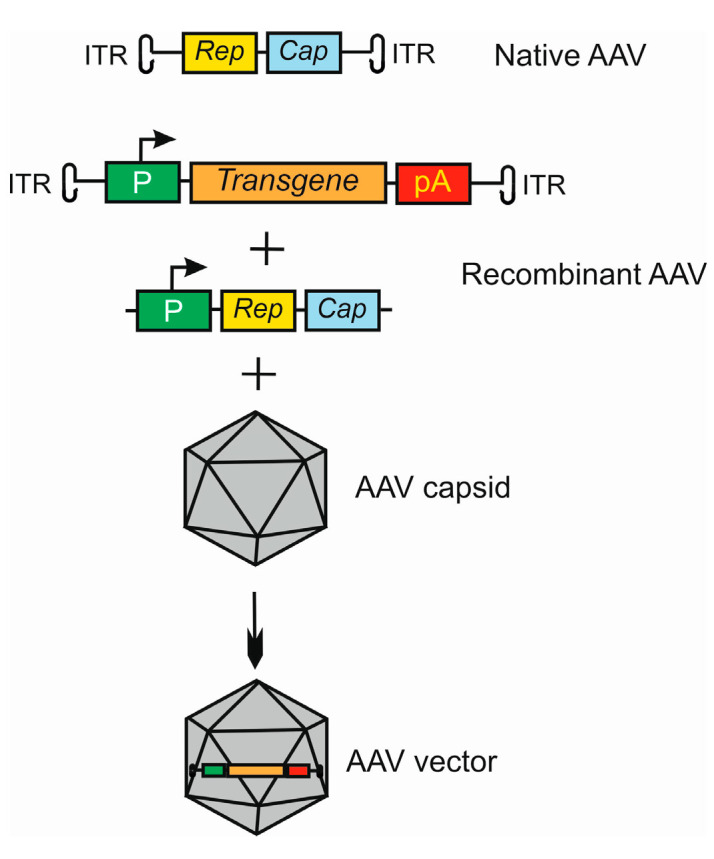
Basic strategy of the use of adeno-associated virus (AAV) as a vector in gene therapy. The AVV genome contains two open reading frames (ORFs), *Rep* and *Cap*, flanked by two inverted terminal repeats (ITRs). Recombinant AAV is formed by replacing ORFs with an expressional cassette containing a promoter (P), and a transgene that may also be an RNA molecule and terminator, here a polyadenylation sequence (pA). The *Rep* and *Cap* are added in *trans.* This construct is packaged into the AAV capsid to form an AAV vector. Many variants of this procedure may be applied, dependent on the tropism and serotype of AAV in conjunction with target cell and tissue type.

**Figure 3 ijms-25-02386-f003:**
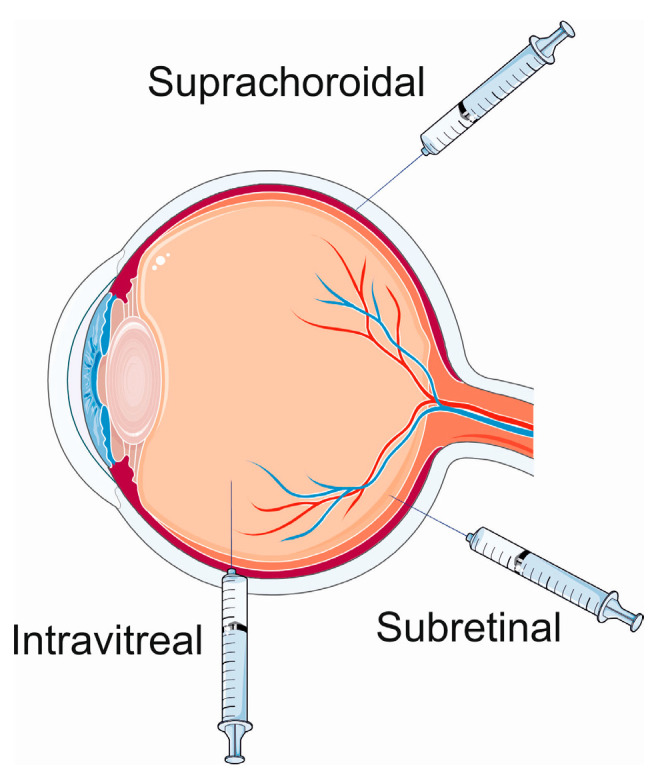
Ocular gene therapy delivery routes. Parts of this figure were drawn by using pictures from Servier Medical Art. Servier Medical Art by Servier is licensed under a Creative Commons Attribution 3.0 Unported License (https://creativecommons.org/licenses/by/3.0/, accessed on 15 January 2024).

**Figure 4 ijms-25-02386-f004:**
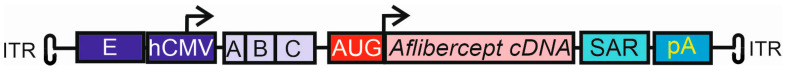
The codon-optimized aflibercept cDNA expression cassette utilized in ADVM-022. The cassette is flanked by adeno-associated virus (AAV) inverted terminal repeats under the control of the human cytomegalovirus early enhancer (E)-promoter (hCMV). These elements are followed by three regulatory sequences of the AAV genome and translation start codon, symbolized here as an RNA codon AUG, within the Kozak consensus sequence and a human scaffold attachment region (SAR) and a polyadenylation termination site (pA) derived from the human growth hormone.

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
