# Peer review of "A New Generation of Gene Therapies as the Future of Wet AMD Treatment"

_ijms, 2024, doi:10.3390/ijms25042386_

Round 1

Reviewer 1 Report

Comments and Suggestions for Authors

In this review article, the authors provide two gene therapy clinical trials ADVM-022, a single injection of ixo[1]vec induced sustained levels of intraocular aflibercept, and RGX-314 , AAV serotype 8 vector in both subretinal and subchoroidal injection of plasmid encoding an anti-VEGF protein, as well as trials with anti-VEGFA treatment-naive patient and gene editing targeting the VEGFA gene to improve wet AMD management. It is well written. However, in the introduction, “These therapies eliminate the burden of multiple intravitreal injections, offering a stable production of anti-VEGF “antibodies” for a long time. Are proteins or antibodies?

Author Response

Comment: In this review article, the authors provide two gene therapy clinical trials ADVM-022, a single injection of ixo[1]vec induced sustained levels of intraocular aflibercept, and RGX-314 , AAV serotype 8 vector in both subretinal and subchoroidal injection of plasmid encoding an anti-VEGF protein, as well as trials with anti-VEGFA treatment-naive patient and gene editing targeting the VEGFA gene to improve wet AMD management. It is well written.

Answer: Thank you.

Comment: However, in the introduction, “These therapies eliminate the burden of multiple intravitreal injections, offering a stable production of anti-VEGF “antibodies” for a long time. Are proteins or antibodies?

Answer: We have replaced “antibodies” with “proteins”

Reviewer 2 Report

Comments and Suggestions for Authors

The authors in this review reported a new approach consisting in gene therapy to replace the gold standard treatment based on the intravitreal injections of anti-VEGF to counteract the neo angiogenesis and the progression of wet AMD.

The authors well described in this review the disadvantages of frequent anti-VEGF injections and conversely the advantages of the gene therapy aimed to induce a continuous production of anti-VEGF inside the eye.

I appreciate the contents of this review, however, I think that should be taken into account that the early event that occurs in the AMD is the dysfunction of retinal pigmented epithelium (RPE) and as a consequence there is a massive release of VEGF and neoangiogenesis. Therefore, this new therapeutic approach could be useful to contrast the neoangiogenesis by inhibiting the effect of VEGF, but in this case the target is represented by a downstream event and not the primary cause of AMD, which is instead to be found in the deregulation of the RPE.

 I invite the authors to extend the discussion adding this important point and highlighting that since this new approach of gene therapy could be powerful for the reasons well explained, a resolutive therapeutic approach for wet AMD should targets RPE maintaining its function at physiological levels.

Author Response

Comment: The authors in this review reported a new approach consisting in gene therapy to replace the gold standard treatment based on the intravitreal injections of anti-VEGF to counteract the neo angiogenesis and the progression of wet AMD.

The authors well described in this review the disadvantages of frequent anti-VEGF injections and conversely the advantages of the gene therapy aimed to induce a continuous production of anti-VEGF inside the eye.

Answer: Thank you.

Comment: I appreciate the contents of this review, however, I think that should be taken into account that the early event that occurs in the AMD is the dysfunction of retinal pigmented epithelium (RPE) and as a consequence there is a massive release of VEGF and neoangiogenesis. Therefore, this new therapeutic approach could be useful to contrast the neoangiogenesis by inhibiting the effect of VEGF, but in this case the target is represented by a downstream event and not the primary cause of AMD, which is instead to be found in the deregulation of the RPE. I invite the authors to extend the discussion adding this important point and highlighting that since this new approach of gene therapy could be powerful for the reasons well explained, a resolutive therapeutic approach for wet AMD should targets RPE maintaining its function at physiological levels.

Answer: We have added the following fragment to Discussion:

“Retinal pigment epithelium is a key structure in AMD pathogenesis, which is especially observed in the dry form of the disease featured by a partial loss of RPE cells. In wet AMD, RPE remains intact, but its dysfunction supports the massive release of VEGFA and neoangiogenesis. Therefore, it must be stressed that new generations of gene therapies revied here are rather a treatment of consequences and not the reason for wet AMD and so they are not an ultimate way of the wet AMD therapy. A further perspective of wet AMD therapy should include AMD-related changes in RPE cells and so it might be universal for both forms of AMD. However, this is a further perspective, and at present several concerns of a new generation of gene therapy still require addressing.”

Round 2

Reviewer 2 Report

Comments and Suggestions for Authors

The RPE is a component of Blood Retinal Barrier and its breakdown is one of the events that occurs during wet AMD, as reported in a recent review (DOI: 10.3390/cells10010064). For this reason I suggest the authors to rephrase this sentence as follow:  "In wet AMD, the breakdown of the Blood Retinal Barrier and the dysfunction of RPE cause the massive release of VEGFA and neoangiogenesis".

Author Response

Comment: The RPE is a component of Blood Retinal Barrier and its breakdown is one of the events that occurs during wet AMD, as reported in a recent review (DOI: 10.3390/cells10010064). For this reason I suggest the authors to rephrase this sentence as follow:  "In wet AMD, the breakdown of the Blood Retinal Barrier and the dysfunction of RPE cause the massive release of VEGFA and neoangiogenesis".

Answer: We have changed the sentence:

“In wet AMD, RPE remains intact, but its dysfunction supports the massive release of VEGFA and neoangiogenesis.”

into

“In wet AMD, the breakdown of the blood-retinal barrier and RPE dysfunction cause the massive release of VEGFA and neoangiogenesis [91].”

with the new reference

  1. Tisi, A.; Feligioni, M.; Passacantando, M.; Ciancaglini, M.; Maccarone, R. The Impact of Oxidative Stress on Blood-Retinal Barrier Physiology in Age-Related Macular Degeneration. Cells 2021, 10, doi:10.3390/cells10010064.